# Influence of Synthesis Conditions on the Properties of Zinc Oxide Obtained in the Presence of Nonionic Structure-Forming Compounds

**DOI:** 10.3390/nano13182537

**Published:** 2023-09-11

**Authors:** Viktor A. Valtsifer, Anastasia V. Sivtseva, Natalia B. Kondrashova, Artem S. Shamsutdinov, Anastasia S. Averkina, Igor V. Valtsifer, Irina N. Feklistova, Vladimir N. Strelnikov

**Affiliations:** 1Institute of Technical Chemistry, Ural Branch, Russian Academy of Sciences, Perm Federal Research Center, Russian Academy of Sciences, 614013 Perm, Russia; valtsifer.v@itcras.ru (V.A.V.); nasthasivtseva@mail.ru (A.V.S.); kondrashova_n_b@mail.ru (N.B.K.); anastasiya.av11@yandex.ru (A.S.A.); valtsifer.i@itcras.ru (I.V.V.); svn@itcras.ru (V.N.S.); 2The Faculty of Biology, Belarusian State University, 220045 Minsk, Belarus; feklistova@bsu.by

**Keywords:** ZnO, hydrothermal, surfactants, morphology, Zn^2+^ ion release

## Abstract

This work investigated the influence of synthesis conditions, including the use of nonionic structure-forming compounds (surfactants) with different molecular weights (400–12,600 g/mol) and various hydrophilic/hydrophobic characteristics, as well as the use of a glass substrate and hydrothermal exposure on the texture and structural properties of ZnO samples. By X-ray analysis, it was determined that the synthesis intermediate in all cases is the compound Zn_5_(OH)_8_(NO_3_)_2_∙2H_2_O. It was shown that thermolysis of this compound at 600 °C, regardless of the physicochemical properties of the surfactants, leads to the formation of ZnO with a wurtzite structure and spherical or oval particles. The particle size increased slightly as the molecular weight and viscosity of the surfactants grew, from 30 nm using Pluronic F-127 (MM = 12,600) to 80 nm using Pluronic L-31 (MM = 1100), PE-block-PEG (MM = 500) and PEG (MM = 400). Holding the pre-washed synthetic intermediates (Zn_5_(OH)_8_(NO_3_)_2_∙2H_2_O) under hydrothermal conditions resulted in the formation of hexagonal ZnO rod crystal structures of various sizes. It was shown that the largest ZnO particles (10–15 μm) were observed in a sample obtained during hydrothermal exposure using Pluronic P-123 (MM = 5800). Atomic adsorption spectroscopy performed comparative quantitative analysis of residual Zn^2+^ ions in the supernatant of ZnO samples with different particle sizes and shapes. It was shown that the residual amount of Zn^2+^ ions was higher in the case of examining ZnO samples which have spherical particles of 30–80 nm. For example, in the supernatant of a ZnO sample that had a particle size of 30 nm, the quantitative content of Zn^2+^ ions was 10.22 mg/L.

## 1. Introduction

Nowadays, metal oxides are extensively used in diverse fields of science and technology. By changing the particle size and shape, it is possible to control the chemical, structural, morphological, mechanical, electrical, optical and other properties of the synthesized material.

It is known that zinc oxide (ZnO) is a promising multifunctional compound broadly utilized in many applications [1,2,3,4,5]. ZnO is used in optoelectronics [6,7], agriculture [8], wastewater treatment [9,10,11] and catalysis [12,13] as a vulcanization activator for rubbers, a hardener or a crosslinking agent for elastomers [14,15], and in the production of electrically conductive carbon materials [16]; it is also used in the cosmetics, medicine and biotechnology industries [17,18,19,20,21], including its use in the elaboration of biodegradable antibacterial packaging [22,23], and photovoltaic solar cells for “green” energy [24,25].

Zinc oxide is also used now as an antibacterial agent against a wide range of bacteria species. It is worth noting that the antibacterial activity of ZnO particles increases when their sizes are reduced to the size range of nanometers [26,27]. However, the mechanism of the antibacterial activity of ZnO particles is still not fully understood. Several hypotheses have been put forward in the literature regarding the probable mechanism of the antibacterial activity of zinc oxide. First of all, direct contact between ZnO nanoparticles or Zn^2+^ ions released from their surface and a bacterial cell wall, and the subsequent violation of its integrity, may be essential. However, it is also possible to assign the antibacterial effect of ZnO to the photocatalytic formation of reactive oxygen species, which generate oxidative stress and irreversibly damage bacterial cells [28,29,30,31,32]. Furthermore, many studies have shown that not only the concentration and size of ZnO particles, but also their morphology [33,34,35] and surface functionalization [36], affect the antibacterial activity of particles. These parameters are determined in turn by synthesis method and conditions, the adjustment of which allows one to obtain the desired textural–structural characteristics of zinc oxide particles [37].

Among the chemical methods for producing nanoparticles, the most common are precipitation from salt solutions [38,39] and the microemulsion method, both using hydrothermal treatment of the reaction mixture and without it [40,41,42]. Precipitation is the simplest and most accessible technique. With this method, zinc oxide particles are formed by reactions of exchange and hydrolysis followed by the subsequent thermolysis of intermediate compounds. Hydrothermal synthesis implies that the mixture of reactants is held at elevated temperatures for a certain time in hermetically sealed high-pressure containers (autoclaves). By varying the various parameters of the process [43], such as the type of precursors, the synthesis and calcination temperature, the pH of the medium, the presence of surfactants and their concentration, it is possible to obtain highly dispersed inorganic compounds of various morphologies. Thus, in the work published in [44], it was shown that the change in the concentration of the surfactant (F127) during the synthesis of cobalt molybdate made it possible to change the morphology of the particles of the compound from nanorods to nanospheres and nanoneedles.

Polyethylene glycols, as well as triblock copolymers (Pluronic), are often used as structure-forming compounds in the process of producing nanoparticles. They can be selectively absorbed at the crystal face of metal oxides and are able to change the growth kinetics of nanoparticles and to act as stabilizers to prevent particle agglomeration [45,46,47,48].

The aim of this study is to investigate the influence of synthesis conditions on the textural–structural properties of zinc oxide obtained via precipitation in the presence of nonionic surfactants of different molecular weights at room temperature and under hydrothermal conditions, including the use of a glass substrate. The effects of synthesized ZnO particle shape and size on the amount of Zn^2+^ ions released from particle surfaces are also analyzed.

## 2. Materials and Methods

In this work, hexahydrate zinc nitrate (Zn(NO_3_)^2^·6H_2_O; AO Vecton, Saint-Petersburg, Russia) was taken as a zinc oxide precursor owing to its good solubility and low decomposition temperature. Aqueous ammonium (NH_4_OH; SIGMATEK GmbH, Landau i. d. Pfalz, Germany) was used as a precipitant. In the synthesis of zinc oxide, nonionic surfactants differing in molecular weights, such as block copolymers of polyethylene and polypropylene oxides (Pluronic F-127, Pluronic P-123, Pluronic L-81, Pluronic L-31; Sigma-Aldrich Chemie GmbH, Taufkirchen, Germany), a block copolymer of polyethylene and polyethylene glycol (PE-block-PEG; Sigma-Aldrich Chemie GmbH, Taufkirchen, Germany), and polyethylene glycol (PEG; Sigma-Aldrich Chemie GmbH, Taufkirchen, Germany) (Table 1) served as structure-forming agents. The surfactants used in the work are polymer compounds with different ratios of hydrophilic (ethylene oxide) and hydrophobic (propylene oxide) constituents. The different affinities for water of parts of the molecules of the compounds gives them the ability to exhibit the properties of micelle-forming surfactants in aqueous solution. The characteristics of the surfactants used in the research are given in Table 1.

The methods used in this study to evaluate the influence of hydrothermal treatment on the textural and structural properties of the zinc oxide samples are as follows: The interaction of Zn(NO_3_)_2_·6H_2_O with NH_4_OH in a medium of nonionic surfactants and further thermolysis of the products of reaction and structure-forming compounds at a temperature of 600 °C—calcination method (C), samples C1–C6, where the number denotes the type of a surfactant according to Table 1;The interaction of Zn(NO_3_)_2_·6H_2_O with NH_4_OH in a medium of nonionic surfactants and aging of washed precipitates under hydrothermal conditions in the presence of a glass substrate—method of hydrothermal synthesis with a glass substrate (HTG), samples HTG1–HTG6;The interaction of Zn(NO_3_)_2_·6H_2_O with NH_4_OH in a medium of nonionic surfactants and aging of the precipitates under hydrothermal conditions in the absence of a glass substrate—HT method, samples HT1–HT6;The calcination of the samples obtained by the HT method at a temperature of 600 °C— HTC method, samples HTC1–HTC6.

The molar ratio of Zn(NO_3_)_2_·6H_2_O and NH_4_OH used in the preparation of zinc oxide samples was [Zn^2+^]/[OH^−^] = 1/2. In all cases, the molar ratio of zinc salts and water was [H_2_O]/[Zn^2+^] = 100. The concentration of nonionic surfactants in the solutions was 0.0085 mol/L.

In the synthesis of zinc oxide samples, the following procedure was used. In the very beginning, we prepared an aqueous surfactant solution at a temperature of 50 °C and sequentially added an aqueous 0.5 M hexahydrate zinc nitrate and ammonium solution to it. Then, the reaction mixture was stirred by a magnetic stirrer (stirring speed—500 rpm) at a temperature of 25 °C for 1 h. The precipitates were separated from the mother liquor by filtration, washed alternatively with ethylene alcohol and distilled water and dried at a temperature of 80 °C. This allowed us to obtain intermediate samples (samples 1–6); each sample was further divided into three parts.

The first part of the precipitates was used to obtain ZnO samples by method C. In this case, the precipitates (already washed and dried) were calcined at a temperature of 600 °C (samples C1–C6).

The second part of the precipitates and the glass substrate were placed in a fluoroplast-lined autoclave, to which distilled water was added, and the mixture was subjected to hydrothermal aging at a temperature of 100 °C for 24 h. Then, the precipitates were removed from the surface of the glass substrate, washed in distilled water to obtain a neutral pH and dried at a temperature of 80 °C (samples HTG1–HTG6).

The third part of the precipitates (no glass substrate) was also held in the autoclave under hydrothermal conditions at a temperature of 100 °C for 24 h. The precipitates were separated from the mother liquor by filtration. The structure-forming compounds were removed by either extraction with ethyl alcohol (samples HT1–HT6) or calcination of the samples at a temperature of 60 °C (samples HTC1–HTC6).

The intermediate compounds and the final zinc oxide samples, which were obtained without hydrothermal treatment of intermediate products and with it, were studied by different physicochemical methods of analysis.

The structure of the intermediate and final products of synthesis was investigated by applying the X-ray diffraction analysis (XRD) and IR spectroscopy techniques. X-ray analysis was performed with an XRD-7000 diffractometer (Shimadzu, Japan) using CuKα-radiation (λcp = 1.54184 Å). Scanning was performed in the angular ranges 2θ = 5–10° and 10–80°. The compounds were identified by comparing with the JSPDS cards. The average crystallite size was determined by the Selyakov–Scherrer formula: = Kλ/(βcosθ), where K = 0.89; λ = 1.54184 Å; β is the half-width of the reflection (100), rad; and θ is the diffraction angle of the reflection. The IR-spectra in the region of 400–4000 cm^−1^ were recorded using a Fourier transform infrared spectrometer IFS–66/S (Bruker, Germany). 

The temperatures needed for the thermal destruction of the intermediate products of synthesis were obtained from thermalgravimetric analysis (TGA) performed with a TGA/DSC 1 (METTLER-TOLEDO, Switzerland) in air at a heating rate of 10 °C/min in the temperature range of 25–1000 °C.

The textural characteristics of the ZnO samples were obtained by the low-temperature nitrogen adsorption method using an ASAP 2020 analyzer (Micromeritics, USA) after outgassing of the material in vacuum at a temperature of 350 °C for 3 h. The specific surface area (SBET) of the samples and the total volume of pores (V_tot_) were obtained by the FRT method, and the pore-size distribution (D_Πop_) in the range of 1.7–300 nm was found from the analysis of desorption isotherms by the BJH method. 

The morphological features of the zinc oxide samples, namely, the average size (DSEM) and shape of particles, were studied by the scanning electron microscopy (SEM) technique using a Quanta FEG650 microscope (Thermo Fisher Scientific, Waltham, MA, USA).

The potential use of the zinc oxide samples synthesized in this study as antibacterial fillers used in polymer composites was evaluated from examining the concentration of zinc ions after the stirring and holding of ZnO in distilled water (C_ZnO_ = 0,062 mol/L) at room temperature for 48 h. Quantitative analysis of the residual Zn^2^+ ions was performed by atomic absorption spectroscopy using an iCE 3500 spectrometer (Thermo Fisher Scientific, Waltham, MA, USA).

## 3. Results

### 3.1. Study of the Structures of Synthesized Products 

X-ray diffraction analysis revealed that, in all cases, the product of the reaction between zinc nitrate and ammonium, regardless of the surfactant used in the synthesis, is hydrous zinc nitrate Zn_5_(NO_3_)_2_(OH)_8_·2H_2_O (JSPDS 24–1460, Figure 1). This process can be represented schematically as:$$5Zn{\left({NO}_{3}\right)}_{2}\cdot 6H_{2}O+8{NH}_{4}OH\to {Zn}_{5}{\left(OH\right)}_{8}{\left({NO}_{3}\right)}_{2}\cdot 2H_{2}O+8{NH}_{4}{NO}_{3}+4H_{2}O.$$

The results of thermal analysis performed on samples 1, 3 and 6 (Figure 2) show that the thermal destruction of the compound Zn_5_(NO_3_)_2_(OH)_8_·2H_2_O is a multi-step process that occurs in the temperature range ~100–300 °C and develops according to the following scheme:(1)$${Zn}_{5}{\left(OH\right)}_{8}{\left({NO}_{3}\right)}_{2}\cdot 2H_{2}O\to {Zn}_{5}{\left(OH\right)}_{8}{\left({NO}_{3}\right)}_{2}+2H_{2}O$$
(2)$${Zn}_{5}{\left(OH\right)}_{8}{\left({NO}_{3}\right)}_{2}\overset{100–170\;{}^{\circ}C}{\to }{Zn}_{3}{\left(OH\right)}_{4}{\left({NO}_{3}\right)}_{2}+2ZnO+2H_{2}O$$
(3)$${4Zn}_{3}{\left(OH\right)}_{4}{\left({NO}_{3}\right)}_{2}\overset{170–220\;{}^{\circ}C}{\to }9ZnO+3Zn{\left({NO}_{3}\right)}_{2}+7H_{2}O+2{HNO}_{3}$$
(4)$$Zn{\left({NO}_{3}\right)}_{2}\overset{220–270\;{}^{\circ}C}{\to }ZnO+{NO}_{2}+NO+O_{2}$$

The thermograms of the other intermediate samples (2, 4 and 5) look similar (Appendix A).

It follows from the TGA data that all structure-forming surfactants used in the synthesis of zinc oxide are completely decomposed in the temperature range 300–600 °C (Figure 2). Thus, the calcination temperature of precipitants (600 °C) was chosen after the samples were obtained by method 1. The weight loss corresponds to the calculated value (~31%) and is slightly more than 30%, which is due to the presence of residual amounts of surfactants.

The formation of synthesized compounds in the presence of nonionic surfactants was studied by IR spectroscopy using samples HT1 and HT5 as examples (Figure 3). The IR spectra of other samples (HT2–HT4 and HT6), regardless of the method of their synthesis, look like those of sample HT1 (Appendix A). Figure 3 shows that a wide band at 3376 cm^−1^ that corresponds to the stretching vibrations of hydroxyl groups can be observed in all cases. A wide wavenumber range and an asymmetry of peaks are caused by the superposition of absorption bands, and this can be attributed to different stretching vibrations. Among them are the stretching vibrations of the free surface OH-groups, the OH-groups connected by a hydrogen bond with oxygen OH∙∙∙O (3400–3700 cm^−1^) and the Zn-vacancy OH-group (3216, 3228 cm^−1^) [49,50,51,52]. This region also has the characteristic absorption bands associated with the presence of the surfactants used in the synthesis of zinc oxide: the ~3300 cm^−1^ band (corresponding to the stretching vibrations of the alcohol hydroxyl and alkoxy groups) overlaps the stretching vibrations of the C–H bonds of alkyl groups.

The sorption of structure-forming compounds on the products of synthesis can be evaluated by analyzing the absorption bands which are characteristic of the surfactants used in this study. For instance, the IR spectra of all samples synthesized with Pluronic-type block polymers have the absorption bands corresponding to the stretching vibrations of the R–CH_3_ (2961 cm^−1^) and R–CH_2_–R (2924 cm^−1^ and 2854 cm^−1^) groups. The samples exhibit the absorption bands of 2918 cm^−1^ νas(CH_2_) and 2849 cm^−1^ νs(CH_2_), which indicate the presence of ethylene fragments. Two intense peaks at 1506 cm^−1^ (1502 cm^−1^) and 1375 cm^−1^, observed in the IR spectra of intermediate compounds of all samples, represent superposition of the absorption bands of the asymmetric and symmetric stretching vibrations of the R–NO_2_ group, as well as the bands indicating the –CH_3_ (1370–1470 cm^−1^) and –CH_2_ (1440–1480 cm^−1^) alkyl groups. Peaks in the wavenumber range of 1117–1046 cm^−1^ can be attributed to the stretching vibrations of the C–O bond in alkoxy groups and primary alcohols. The bands of ~800 and ~700 cm^−1^ correspond to the deformation vibrations of the –CH_3_ and –CH_2_ groups. These bands can be assigned, according to some data, to the vibrations of Zn–O bonds on the surface of crystals, where the Zn^2+^ cation is coordinatively unsaturated (~800 cm^−1^) [49,50], as well as to the multiphonon lattice vibrations (~700 cm^−1^) [51], or the bond deformation vibrations of Zn–O–H (833–850 cm^−1^) [53]. 

After the structure-forming compounds were removed by extraction with ethyl alcohol or via calcination at a temperature of 600 °C, the IR spectra of all samples (with the exception of HTG1 and HT1), showed the formation of one intense absorption band at ~500 cm^−1^ corresponding to the stretching vibrations of the Zn–O bond (Figure 3, sample HT5). The IR spectra of the HTG1 and HT1 samples produced under hydrothermal conditions using the high-molecular surfactant Pluronic F-127 show the adsorption bands corresponding to hydroxocarbonate structures: ~3300, 1628 cm^−1^ and a wide intense band at 1046 cm^−1^ (Figure 3, sample HT1) [54,55].

The IR-spectroscopy data are consistent with the X-ray diffraction analysis data (Figure 4). It is shown that the crystal structure of almost all investigated compounds corresponds, regardless of the type of synthesis technique, to the structure of zincite (JSPDS 36–1451, spatial group P63mc). Only in the HTG1 and HT1 samples, which were obtained by methods 2 and 3 in the presence of Pluronic F-127, in addition to the zincite structure, the zinc hydroxocarbonate phase Zn_5_(CO_3_)_2_(OH)_6_ (JSPDS 19–1458, spatial group C2/m) was determined. The formation of this compound can be attributed to the fact that, although Pluronic F-127 has a relatively low thermal destruction temperature (180–350 °C, Figure 2) and a high content of hydrocarbon fragments, it decomposes under excess pressure and a large amount of carbon dioxide is released. The hermetic reactor prevents the release of CO_2_ into the atmosphere, contributing to its dissolution in distilled water and promoting the formation of zinc hydroxocarbonate structures.

The crystallite sizes in the HTG1 and HT1 samples obtained in the presence of Pluronic F-127, which has the highest molecular weight of all surfactants considered in the work, are the smallest among the compounds prepared using Pluronics and are ~26 nm and ~17 nm, respectively. (Table 2). 

Analysis of the data given in Table 2 indicates that the crystallite size in the zinc oxide samples, obtained during hydrothermal aging (methods HTG, HT and HTC) and without it (method C), mostly grows with a decrease in the mean molecular weight of the surfactants used in the ZnO synthesis. It is worth noting that the crystallite size in the zinc oxide samples obtained in conditions of hydrothermal aging and further removal of surfactant by ethyl alcohol extraction (HTG and HT) is somewhat smaller compared to the samples obtained by the methods in which surfactants were removed via calcination (methods C and HTC).

The textural properties of synthesized compounds were investigated using the low-temperature nitrogen sorption technique. It was found that the sorption curves plotted for all samples are one-type curves (Figure 5, Appendix A). Figure 5 presents sorption isotherms and pore-size distribution curves for some ZnO samples. Table 2 contains the textural–structural characteristics of the ZnO samples obtained by different methods. The data demonstrate that the sorption isotherms of all samples can be assigned to type II (IUPAC). The rise (on the left) of the pore-size distribution curves and almost horizontal part of the sorption curves are an indication of micropores. A wide rise of the pore distribution curves within the interval of 4–40 nm is most likely associated with interparticle distances. Generally, the pore-size distribution curves confirm their “not one dimensional” character. 

Table 2 shows that the volume of sorbed nitrogen and the specific surface area in the samples obtained by the methods where the excess structure-forming compounds were removed by ethyl alcohol extraction are higher than in the samples in which the surfactant was removed by calcination at a temperature of 600 °C. The HTG1 and HT1 samples (118 m^2^/g and 145.48 m^2^/g), which were synthesized in the presence of high molecular Pluronic F-127 and the composition of which involves, along with the zincite phase, the zinc hydroxycarbonate phase, have the greatest specific surface area. It is evident that the high textural characteristics of these samples are determined by the morphology of their surface.

### 3.2. The Morphological Features of Synthesized ZnO Samples 

The morphological features of the synthesized ZnO samples, which were investigated using the scanning electron microscopy technique, are illustrated in Figure 6, Figure 7 and Figure 8.

It is seen from Figure 6 that the zinc oxide samples C1–C6 are composed of particles that are spherical and oval in shape. The particle size in the C1 and C2 samples, obtained in the presence of high-molecular surfactants (Pluronic F-127 and P-123), are significantly smaller compared to other samples and are equal to ~30 and ~40 nm, respectively. Analysis of the micrographs of these samples revealed crystalline structures in the form of hexagonal pyramids, which occur already at room temperature over a short time period (for 1 h). It can also be observed that the particle size in the zinc oxide samples produced by method C in the presence of Pluronic structure-forming compounds somewhat increases as the molecular weight of the surfactants used decreases. 

The particle morphology of the zinc oxide samples HTG1–HTG6 and HT1–HT6, which were synthesized under hydrothermal conditions, differs significantly from the particle shape in the samples obtained by method C (Figure 7 and Figure 8). 

The formation of particles of different shapes can be explained using the experimental data collected from the literature as follows.

IR spectroscopy demonstrated that the surfactants dissolved in water covered the formed intermediate compound Zn_5_(NO_3_)_2_(OH)_8_·2H_2_O crystals, whose growth is limited by the residence time of these crystals in the reactor and by the reaction medium temperature (in our case, 1 h at a temperature of 25 °C). When zinc oxide is obtained by method C, the surfactant adsorbed on the particles is removed via calcination at the temperature determined by TGA—600 °C. The shape of the zinc oxide particles is spherical or oval in this case.

The mandatory step in synthesizing the samples by methods HTG and HT under hydrothermal conditions includes the separation of the intermediate compound Zn_5_(NO_3_)_2_(OH)_8_·2H_2_O as precipitate from the mother liquor and its thorough washing, first with alcohol, and then with water. In the case of the HTG5 and HT5 samples, due to the high content of the PE-block-PEG polyethylene component in the surfactant, in order to realize this goal, it is also desirable to use a nonpolar solvent. After this step, the washed precipitates placed in distilled water are hydrolyzed under hydrothermal conditions and Zn(OH)_4_^2−^ ions are formed. The subsequent dehydration of these precipitates causes the formation of zinc oxide [56]: $${{Zn(OH)}_{4}}^{2-}=ZnO+H_{2}O+2{OH}^{-}$$

Zinc oxide crystallizes in the hexagonal structure of zincite, where each atom of a single element is surrounded by four atoms of another element located at the vertices of the tetrahedron (Figure 9) [2,57].

Since ZnO is a polar crystal composed of polar planes ending in zinc and oxygen, the growth rates of crystal planes are different in the presence of compounds with different polarities. 

According to the data summarized in Table 1, block polymers 1–4 used in this study as structure-forming compounds contain a hydrophobic polypropylene oxide (PPO) block and a hydrophilic polyethylene oxide (PEO) block. Compound 5, PE-block-PEG, also has two parts: polar and nonpolar. The ratio of the molecular weights of polar and nonpolar surfactant components, the so-called lipophilic balance, primarily determines the size and shape of the formed crystalline particles. 

Of all the structure-forming compounds considered here, PEG (MM = 400) is the most polar and hydrophilic material. Therefore, the use of PEG in the synthesis of ZnO promotes the growth of long spear-shaped structures, which are often assembled into crystal intergrowths (Figure 7 and Figure 8). 

Based on the experimental data, we can conclude that the molecular weight and viscosity of the Pluronic-type structure-forming compounds have a significant effect on the size and shape of zinc oxide particles.

In particular, it can be seen from Figure 8 that the largest crystalline structures >10–20 µm are formed when zinc oxide is produced by the HT method and when Pluronic P-123 (molecular weight 5800, Table 1) is used as a structure-forming compound. Star-like ZnO crystal intergrowths were detected in this sample as well.

It is worth noting here that the presence of a glass substrate in the hydrothermal aging process performed to obtain zinc oxide samples by the HTG method provides the formation of smaller (500–600 nm) one-dimensional crystal structures (Figure 7), and its lack (the HT method) contributes to the uncontrolled growth of particles, the sizes of which differ from each other by several times (Figure 8). 

It can be seen from Figure 7 and Figure 8 that the HT1 and HTG1 samples, produced in the presence of surfactants with the highest molecular weight (12,600), exhibit a morphology that differs from that of other samples obtained by the same methods. The hydrothermal aging results in the formation of flaky fine structures collected into loose aggregates. As shown before, this is due to the fact that along with the ZnO phase, these samples also contain the zinc hydroxocarbonate phase Zn_5_(CO_3_)_2_(OH)_6_. When the HT1 sample is calcined at a temperature of 600 °C, zinc hydroxocarbonate is destroyed. Large crystalline ZnO structures with a size of ~10 µm can be observed in the HTC sample (Figure 10). Note that the calcination of other samples of this series at a temperature of 600 °C had almost no effect on the particle size and shape.

### 3.3. Determination of Residual Zn^2+^ Ions in the Supernatant of Synthesized Samples

Zinc oxide is actively used now as an antibacterial agent against a wide range of bacteria. One of the most common mechanisms of antibacterial activity is the contact between the Zn^2+^ ions released from the particle surface and the bacterial cell wall, and the subsequent violation of the cell’s integrity.

The antibacterial activity of ZnO nanoparticles was studied in [58] against *B. subtilis*, *E. coli* and *Pseudomonas fluorescens* bacteria. It was shown that ZnO nanoparticles (about 70 nm in size) with a concentration of 3 to 12 mmol/L caused 100% mortality of the studied bacteria. The work discussed in [59] determined that increasing the concentration of zinc ions improved the growth inhibitory effect of *E. coli*. There are also studies supporting the high antimicrobial activity of ZnO particles against microorganisms such as *Salmonella typhimurium* and *Staphylococcus aureus* [22].

Based on the literature data, this work assessed the potential use of synthesized ZnO samples in the creation of bactericidal materials.

The application of the atomic absorption spectroscopy technique made it possible to perform a quantitative analysis of the supernatant solutions for residual Zn^2+^ ions after holding the ZnO samples, obtained by the C and HTG methods, in distilled water at room temperature and after 48 h of continuous stirring. The obtained results are given in Table 3.

It follows from the data of Table 3 that the residual amount of Zn^2+^ ions is expected to be higher if the supernatant of the ZnO samples obtained by the C method, which have spherical particles of 30–80 nm, is examined for this purpose.

## 4. Conclusions

In this work, the influence of synthesis conditions, including the application of structure-forming compounds of different molecular weights and the use of a glass substrate in the hydrothermal aging process, on the textural–structural properties of zinc oxide samples was evaluated. 

The X-ray diffraction analysis indicated that when hydrous zinc nitrate (used as a ZnO precursor) reacts with the ammonium solution in a medium of nonionic surfactants, Zn_5_(OH)_8_(NO_3_)_2_·2H_2_O is formed as an intermediate compound.

The IR spectroscopy technique revealed that the precipitates of the intermediate compound Zn_5_(NO_3_)_2_(OH)_8_·2H_2_O (pre-washed with alcohol and distilled water) have on their surface functional fragments corresponding to the surfactants used in the ZnO synthesis.

The obtained data demonstrated that the hydrothermal aging of samples at 100 °C causes the crystal ZnO structures to appear, and the use of a glass substrate in this case contributes to the formation of one-dimensional hexagonal particles with size of 0.6–1.5 μm. The absence of a glass substrate in hydrothermal aging leads to a significant growth of zinc oxide particles of different sizes. It is shown that the largest particles were observed in the sample obtained during hydrothermal aging without a glass substrate and in the presence of Pluronic P-123 (MM = 5800) with an ethylene dioxide content of 30% by weight.

It was found with the help of X-ray diffraction analysis that the samples obtained by hydrothermal treatment where the high molecular Pluronic F-127 was used as a structure-forming compound involve, in addition to the ZnO phase, the zinc hydroxycarbonate phase Zn_5_(CO_3_)_2_(OH)_6_.

The SEM technique revealed that the holding of the structured zinc oxide samples at 600 °C in the presence of surfactants with a molecular weight under hydrothermal conditions has practically no effect on the size and shape of the formed hexagonal particles.

The quantitative analysis of residual Zn^2+^ ions in the supernatant of ZnO samples was performed by the atomic absorption spectroscopy method. It was established that the Zn^2+^ content in the studied liquids is higher in the case of two days of exposure to zinc oxide samples that have spherical and oval particle shapes of 30–80 nm compared to ZnO samples that have structured particles of 0.6–1.5 μm. 

Therefore, nanoscale ZnO particles obtained by the C method can be further considered in the development of bactericidal materials.

## Figures and Tables

**Figure 1 nanomaterials-13-02537-f001:**
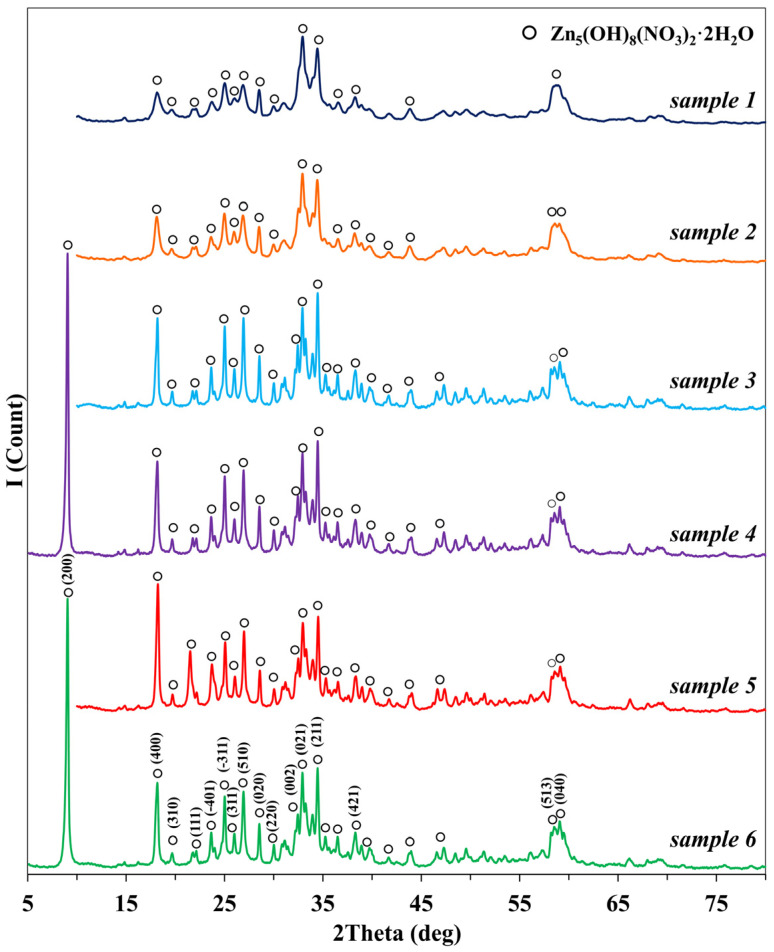
X-ray diffraction patterns of the intermediate products obtained in the synthesis of zinc oxide (samples 1–6).

**Figure 2 nanomaterials-13-02537-f002:**
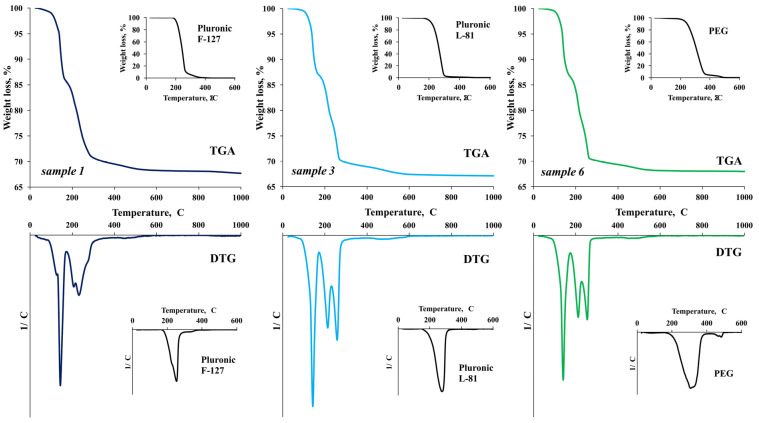
Data illustrating the thermal destruction of Zn_5_(NO_3_)_2_(OH)_8_·2H_2_O.

**Figure 3 nanomaterials-13-02537-f003:**
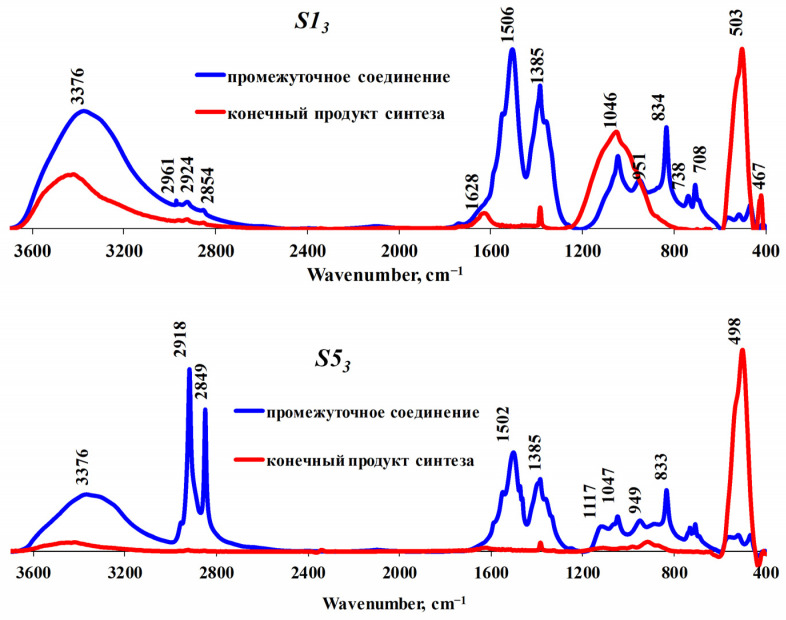
IR spectra of zinc oxide samples synthesized in the presence of different surfactants.

**Figure 4 nanomaterials-13-02537-f004:**
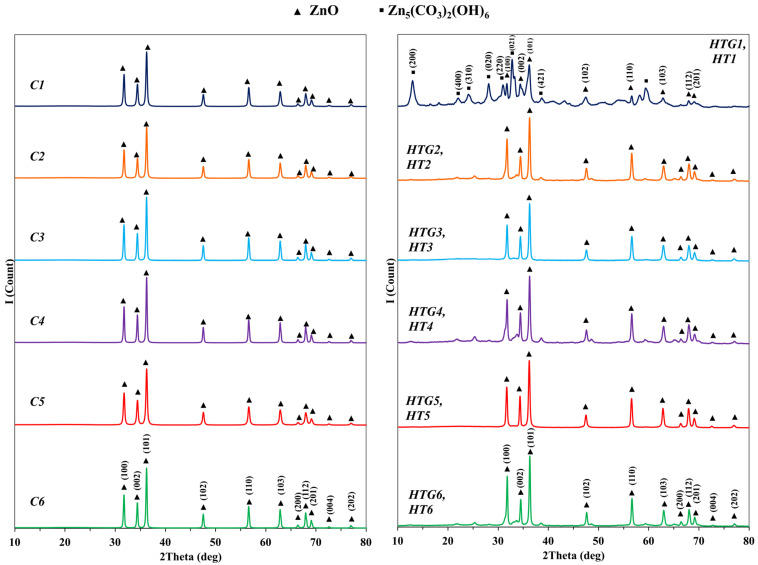
X-ray diffraction patterns of the zinc oxide samples obtained in the presence of nonionic surfactants by the *C* method (at left) and by the HTG and HT methods (at right) under hydrothermal conditions.

**Figure 5 nanomaterials-13-02537-f005:**
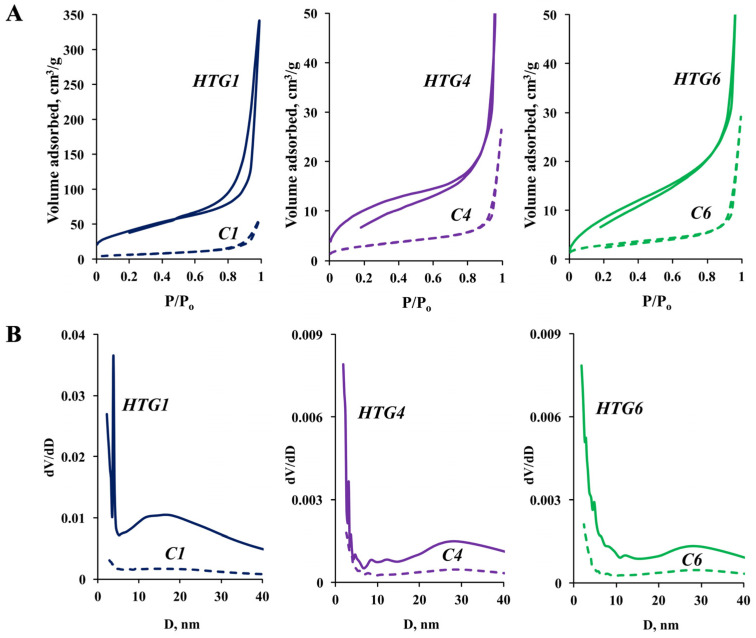
Sorption isotherms (**A**) and pore-size distribution curves (**B**) for synthesized samples.

**Figure 6 nanomaterials-13-02537-f006:**
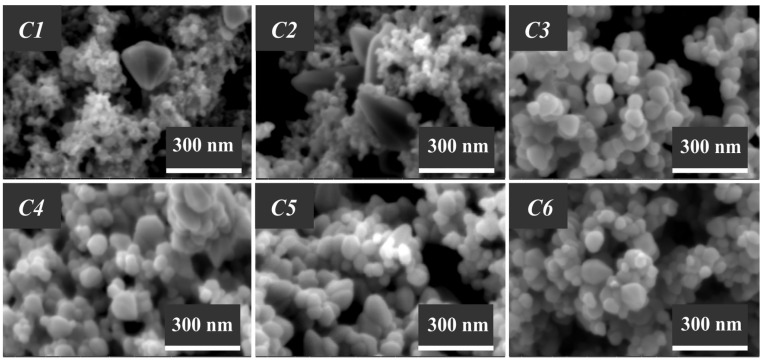
Micrographs of the zinc oxide samples (C1–C6) produced by the C method.

**Figure 7 nanomaterials-13-02537-f007:**
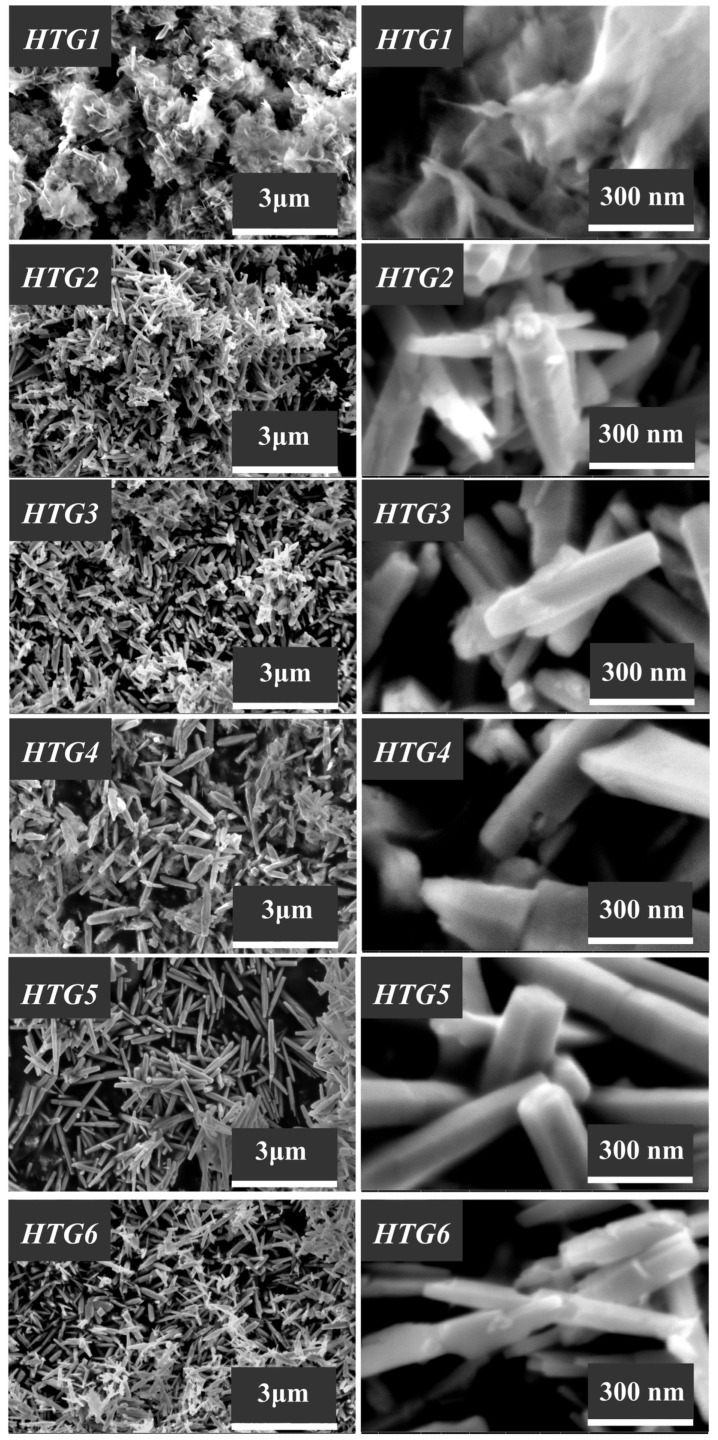
SEM images of the zinc oxide samples (HTG1–HTG6) obtained by the HTG method.

**Figure 8 nanomaterials-13-02537-f008:**
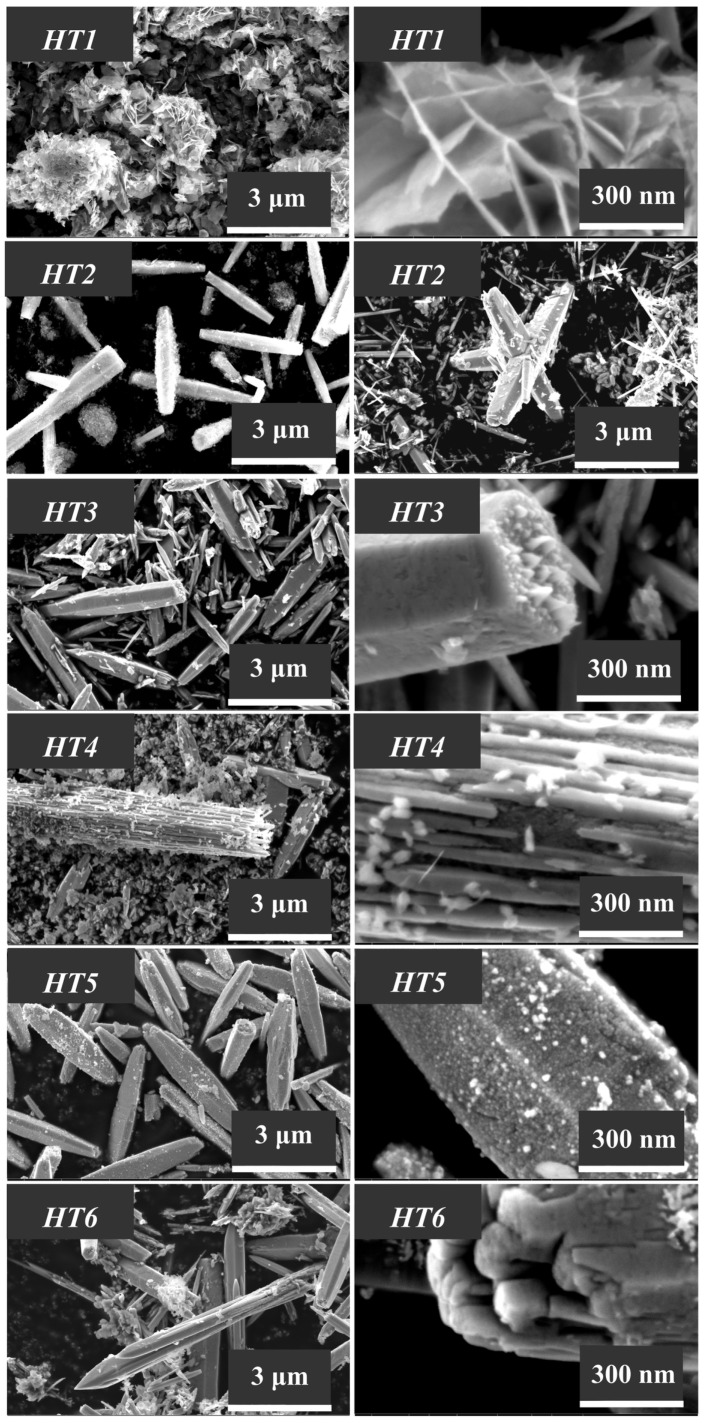
SEM images of the zinc oxide samples (HT1–HT6) obtained by the HT method.

**Figure 9 nanomaterials-13-02537-f009:**
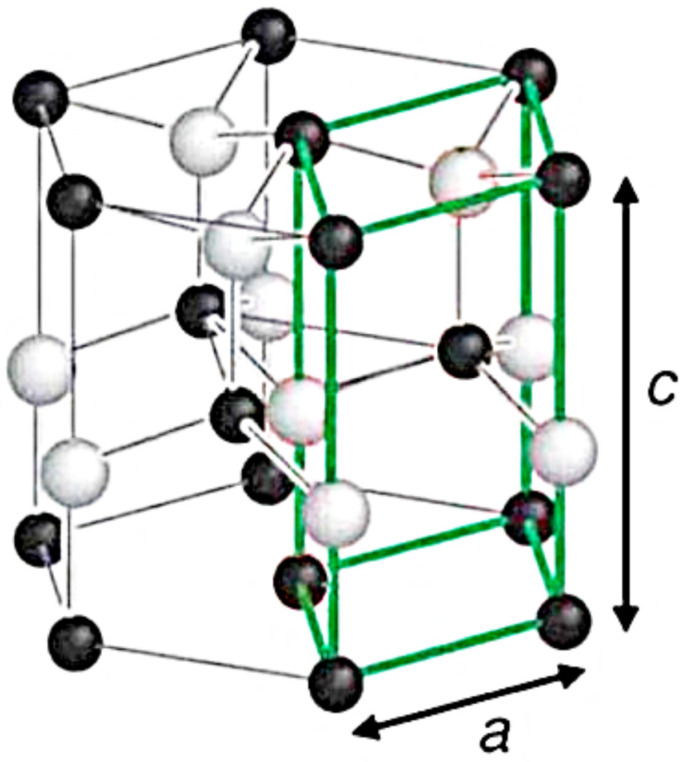
Schematic diagram of the hexagonal structure of ZnO wurtzite with lattice constants *a* and *c*. The dark and white spheres are zinc and oxygen ions, respectively. The primitive shell is shown by green lines [2,57].

**Figure 10 nanomaterials-13-02537-f010:**
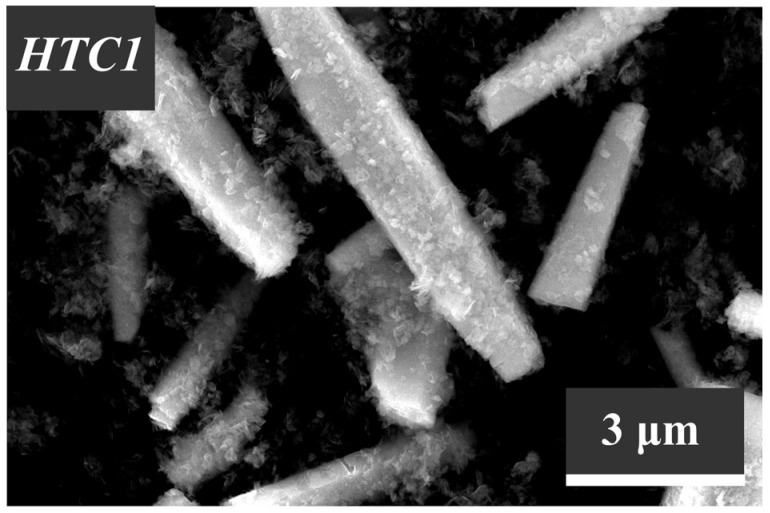
SEM image of the HTC1 sample.

**Table 1 nanomaterials-13-02537-t001:** The main characteristics of the structure-forming compounds used in the synthesis of zinc oxide.

№	Name	Formula	Mean Molecular Weight (MW), g/mol	Ethylene Oxide Concentration, Weight. %	Surfactant Solution Viscosity (C = 0.0085 mol/L), mPa·s
0	Distilled water	H_2_O	18	–	0.88
1	Pluronic F-127	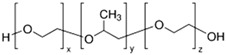	12,600	70	5.95
2	Pluronic P-123		5800	30	1.60
3	Pluronic L-81		2800	10	1.17
4	Pluronic L-31		1100	10	1.12
5	PE-block-PEG	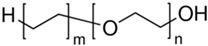	575	20	1.19
6	PEG	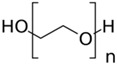	400	100	1.02

**Table 2 nanomaterials-13-02537-t002:** The textural–structural characteristics of synthesized samples.

Synthesis Method	Surfactant Removal Technique	Sample	Specific Surface Area, *S_BET_*, m^2^/g	Total Pore Volume, *V_tot_*, cm^3^/g	Pore Diameter, *D_pore_*, nm	Crystallite Size, nm (XRD)
C	Calcination at 600 °C	C1	22.84	0.088	14.67	28.25
		C2	18.79	0.087	17.61	31.31
		C3	13.23	0.067	17.38	31.80
		C4	10.33	0.045	18.03	34.69
		C5	8.56	0.030	16.56	35.52
		C6	9.82	0.041	18.75	35.87
HTG	Ethyl alcohol extraction	HTG1	117.79	0.605	20.542	25.99
		HTG2	19.16	0.102	21.207	26.39
		HTG3	10.44	0.043	16.636	27.34
		HTG4	75.09	0.382	20.326	29.46
		HTG5	8.25	0.027	12.964	32.24
		HTG6	11.54	0.030	10.372	30.65-
HT	Ethyl alcohol extraction	HT1	145.48	0.519	13.60	16.87
		HT2	24.79	0.071	16.20	27.10
		HT3	21.11	0.072	15.39	28.03
		HT4	35.69	0.202	23.00	28.19
		HT5	38.81	0.258	23.59	27.46
		HT6	34.73	0.138	14.25	33.36
HTC	Calcination at 600 °C	HTC1	24	0.09	15	33.92
		HTC2	24	0.10	14	34.84
		HTC3	10	0.04	14	35.76
		HTC4	10	0.04	14	36.18
		HTC5	6	0.015	12	37.96
		HTC6	15	0.05	13	33.42

**Table 3 nanomaterials-13-02537-t003:** The residual amounts of zinc ions after holding the ZnO samples synthesized by the *C* and *HTG* methods in distilled water.

Synthesis Method	Sample	Mean Particle Size, μm (SEM)	Residual Amount of Zn^2+^ Ions, mg/L
C	C1	0.03	10.22
	C2	0.04	9.18
	C3	0.07	8.50
	C4	0.08	8.06
	C5	0.08	7.86
	C6	0.08	7.94
HTG	HTG1	2.0	8.69
	HTG2	0.6	6.64
	HTG3	0.6	6.86
	HTG4	1.5	5.04
	HTG5	1.5	5.38
	HTG6	1.5	5.75

## Data Availability

Data will be made available on request.

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
