# Peer review of "Influence of Synthesis Conditions on the Properties of Zinc Oxide Obtained in the Presence of Nonionic Structure-Forming Compounds"

_nanomaterials, 2023, doi:10.3390/nano13182537_

Round 1
Reviewer 1 Report
I have reviewed the submitted manuscript entitled “Influence of Synthesis Conditions on the Properties of Zinc Oxide Obtained in the Presence of Non-ionic Structure-Forming Compounds” by V. A. Valtsifer et al submitted to Nanomaterials/MDPI. The authors have addressed the formation and properties of ZnO in dependence on the hydrothermal synthesis procedures. The chemical and microstructural properties of ZnO particles are important factors in these applications and for that reason, different methods were used in the synthesis of ZnO particles. Generally, by changing the synthesis methods it is possible to change the properties of metal oxide particles. Surfactants play an important role in the morphology and particle size of ZnO. The highest antibacterial activity was shown for the sample synthesized in the supernatant solution obtained by the C and HTG methods.
The manuscript is well-drafted, and some minor modifications are required.
· The ionic and non-ionic surfactants must be spelled out rather than leaving it very general.
· How the microemulsion method compares to that of a hydrothermal method for synthesizing ZnO?
· How the surfactants-controlled formation of small ZnO particles without any agglomeration.
· Please compare the results reported in the literature (such as doi.org/10.1002/cplu.201600294) for varying the concentration of surfactant (F127) varies the morphology from nanorods to nanospheres and nanoneedles.
· Why the XPD peaks are so poor with a large peak-to-noise ratio?
· How the samples1 - 6 differ?
· Are the OR spectra averaged for multiple scans?
· The CH3 and CH2 groups shown in the IR spectra must be referred to the literature such as (10.1039/D0DT01871F)
· Please provide the surface area of the samples.
· Please draw a succinct conclusion and summarise the SEM images with varying particle sizes and shapes obtained by the HTG method.
It is fine.
Reviewer 2 Report
The manuscript of Viktor A. Valtsifer and colleagues explored the relationship between the textural-structural properties and synthesis conditions of zinc oxide samples. First of all, during the synthesis process of zinc oxide samples, various nonionic surfactants differing in molecular weights were screened. In-depth functional analysis of samples by using X-ray diffraction, TGA, IR and SEM techniques. The residual amount of zinc oxide ions after the ZnO samples synthesized by different methods were kept in distilled water was compared. In a word, this study fits with the scope of the journal. However, the manuscript needs major revisions before possible publication in nanomaterials. The comments are provided as below:
1. In the description of line 90, the calcination temperature of method C is 600 ℃, but in the description of line 114, it is 60 ℃. why?
2. Contents of lines 116-120 and lines 121-125 are completely repeated, please check and revise them carefully.
3. In Table 2, the crystallite size of HTG1 is 25.99, but the description of the paragraph at lines 239-242 is inconsistent with the experimental data. Please rewrite the paragraph to ensure the scientific expression.
4. The author is requested to supplement the experiment for explaining the relationship between different zinc oxide ions residues and antibacterial properties.
5. Some format problems could be found in the whole manuscript. Please carefully revised to meet the requirements of the journal:
a. In line 13, “It is shown that that the zinc oxide particles produced by...” should be changed to “It is shown that the zinc oxide particles produced by...”.
b. In line 168, “range ~120 - 300 ℃” should be revised to “range 100 - 300 ℃”, according to the reaction formula of line 170-173.
c. In line 95, in Figure 4, “HTG1-HTG2” should be certified whether is “HTG1-HTG6” nor not.
d. In the section of References, some document formats are inconsistent, such as Ref.7, Ref.8 and Ref.19. Please check carefully and unify the reference format, which meet the requirements of the journal.
e. The headers and footers of Tables 1-3 should write as bold, that can make them look clearer visually.
Minor editing of English language required.
Reviewer 3 Report
The article "Influence of Synthesis Conditions on the Properties of Zinc Oxide Obtained in the Presence of Nonionic Structure-Forming Compounds" describes the synthesis and characterization of ZnO nanoparticles by employing various starting conditions. It is a valuable study that can be published after authors address the following problems:
Abstract should be checked and revised carefully by briefly introducing the work plan and key findings. Abstracts should highlight the innovation of the article, as often abstract section is presented separately in search engines, it must be able to stand alone as an informative piece. In the abstract, need to focus more on the quantitative information, not qualitative one.
Use ZnO as keyword instead of zinc oxide (it is already in the title) – will improve hit chances in searches. In addition, I recommend using hydrothermal, surfactants and morphology as keywords. As antibacterial properties were not determined in any way that keyword should be eliminated- it can be replaced with “Zn2+ ions release”.
This work is interesting and can be boosted further. Following literature could prove this manuscript: doi: 10.3390/pharmaceutics14122842; doi: 10.3390/ijms24065677; doi: 10.3390/ma16155400; doi: 10.3390/ijms23179541; doi: 10.3390/ma16083275 as similar factors influencing the synthesis, antibacterial activity and intermediates are reported - Zn5(CO3)2(OH)6 and Zn5(OH)8(CH3OO)2.
In addition, under introduction, at rows 32-36 please update with food packaging and other important energy related applications e.g. doi: 10.3390/pharmaceutics13071020; doi: 10.3390/ijms23095218; doi: 10.3390/ma16134510; doi: 10.3390/ma16124297; doi: 10.3390/ijms23148022.
At rows 95, 120 should be HTG1-HTG6.
The English language needs some polishing for style and typos (e.g. zink – rows 102, 126; use IUPAC names as hexahydrate instead of 6-aqueous; precipitants –row 179; missing verb at row 180 - were “obtained” by method 1; row 184 Cyrillic letter; row 344 “cryslalline”.
At row 114 I assume it should be 600oC as in row 90.
Rows 121-125 should be deleted as they repeat rows 116-120.
Why authors used λ=1,54056 Å in Scherrer formula?
Please provide scanning speed for XRD and FTIR. How many spectra were averaged in FTIR and with what resolution.
At row 162 please correct the reaction equation, as NH4NO3 instead separate NH3 and HNO3 – you cannot hope to obtain separate base and acid in the same solution.
In my opinion as authors have extensively presented the XRD results for each sample, they can add in a Supplementary file the TG, FTIR, BET and SEM results for all samples (for SEM only rest of HTC). Adding these results will only improve the quality of the paper and attract a broader readership.
Figure 7 caption should be HTG like in micrographs. Figure 8 caption should be HT like in micrographs.
Authors should check the correct variant of the above figures as in the text reference, rows 349-352, they state that smaller structure are formed in HTG method, but reference the Figure 8 (while in fact the smaller crystals are in Figure 7). Same problem with HT which is referenced as Figure 7. In this respect if the Figure 7 is HTG and Figure 8 is HT, then at row 353 the sentence should be switched as “HTG1 and HT1”.
Use uniform notation for measurement units (now for litre are used both l and L like in Table 1 and Table 3 but also elsewhere across the manuscript). Personally, I would recommend the use of L.
For Table 3 results in which way size of crystallite and size of particles influence the amount of Zn2+ ions released into the water?
The conclusion should reflect the heuristic of the study. Motivation must be clear and authors should better explain the reasons behind chosen this system and their findings in a critical way.
My suggestion for authors is to follow up this study with practical applications like photocatalysis and antimicrobial activity determinations.
The English language needs some polishing for style and typos (e.g. zink – rows 102, 126; use IUPAC names as hexahydrate instead of 6-aqueous; precipitants –row 179; missing verb at row 180 - were “obtained” by method 1; row 184 Cyrillic letter; row 344 “cryslalline”.
Round 2
Reviewer 3 Report
Authors have responded to most of my concerns. The issue below can be corrected in proofing phase as it does not influence significantly the results.
The answer to issue 9 "In the Scherrer formula, the radiation wavelength λsr = 1.54184 Å corresponds to the characteristics of the XRD-7000 diffractometer instrument (Shimadzu, Japan), where CuKα emission is used." I can not find it solved - the problem is that three rows above authors declare (λср = 1.54184 Å) while in Scherrer formula use λ=1,54056 Å.